# Comparison of the Effect of Corn-fermented Protein and Traditional Ingredients on the Fecal Microbiota of Dogs

**DOI:** 10.3390/vetsci10090553

**Published:** 2023-09-02

**Authors:** Logan R. Kilburn-Kappeler, Tyler Doerksen, Andrea Lu, Rachel M. Palinski, Nanyan Lu, Charles G. Aldrich

**Affiliations:** 1Department of Grain Science and Industry, Kansas State University, Manhattan, KS 66506, USA; lkilburn@ksu.edu; 2Veterinary Diagnostic Laboratory, College of Veterinary Medicine, Kansas State University, Manhattan, KS 66506, USA; tdoerks@vet.k-state.edu (T.D.); andrealu@vet.k-state.edu (A.L.); rachel.palinski@usda.gov (R.M.P.); 3Department of Diagnostic Medicine/Pathobiology, College of Veterinary Medicine, Kansas State University, Manhattan, KS 66506, USA; 4Bioinformatics Center, Kansas State University, Manhattan, KS 66506, USA; nanyan@ksu.edu

**Keywords:** canine diets, corn-fermented protein, dried yeast, 16S metagenomic sequencing, fecal microbiota

## Abstract

**Simple Summary:**

Corn-fermented protein, a co-product of ethanol production, can be utilized as a protein source for pet food. Currently, there are no studies that have evaluated the impact of this ingredient on the fecal microbiota of dogs, an indicator of animal health. The overall richness and diversity of the fecal microbiota were maintained when dogs were fed corn-fermented protein compared to traditional ingredients such as brewer’s dried yeast and distiller’s dried grains with solubles.

**Abstract:**

Corn-fermented protein (CFP), a co-product from the ethanol industry, is produced using post-fermentation technology to split the protein and yeast from fiber prior to drying. The objective of this study was to determine the effect of CFP compared to traditional ingredients on the fecal microbiota of dogs. The four experimental diets included a control with no yeast and diets containing either 3.5% brewer’s dried yeast, 2.5% brewer’s dried yeast plus 17.5% distiller’s dried grains with solubles, or 17.5% CFP. The experimental diets were fed to adult dogs (*n* = 12) in a 4 × 4 replicated Latin square design. Fresh fecal samples (*n* = 48) were analyzed by 16S metagenomic sequencing. Raw sequences were processed through mothur. Community diversity was evaluated in R. Relative abundance data were analyzed within the 50 most abundant operational taxonomic units using a mixed model of SAS. Alpha and beta diversity were similar for all treatments. Predominant phyla among all samples were Firmicutes (73%), Bacteroidetes (15%), Fusobacteria (8%), and Actinobacteria (4%). There were no quantifiable (*p* > 0.05) shifts in the predominant phyla among the treatments. However, nine genera resulted in differences in relative abundance among the treatments. These data indicate that compared to traditional ingredients, CFP did not alter the overall diversity of the fecal microbiota of healthy adult dogs over 14 days.

## 1. Introduction

Diet has a tremendous impact on the composition and function of the fecal microbiota. Dietary fiber and other non-digestible carbohydrates are the most studied and are well known to benefit the gastrointestinal microbiota [1,2,3,4,5,6]. Dietary fiber can increase the production of short-chain fatty acids (SCFAs) and lactate in the intestine, which provide sources of energy throughout the body, contribute to intestinal function, and prevent disease. Dietary fiber also increases microbial metabolic activity and the Firmicutes phylum [7,8,9]. Therefore, increased production of SCFAs and the relative abundance of Firmicutes due to dietary intervention would be desirable. In addition to dietary fiber, the amount and type of dietary protein may impact the fecal microbiota. Dietary protein can increase putrefactive compounds such as ammonia, indoles, and amines due to amino acid degradation by the intestinal microbiota. Dietary protein has also been reported to increase bacteria belonging to the Fusobacteria and Proteobacteria phyla [10]. Therefore, a decrease in putrefactive compounds and the relative abundance of Fusobacteria and Proteobacteria due to dietary intervention would be beneficial.

Research evaluating the impact of dietary intervention on the canine microbiome is relatively new. Therefore, the effect of specific ingredients is largely unknown. Further research in this field could provide additional direction on how to modulate diet to promote animal health.

Distiller’s dried grains with (DDGS) or without solubles (DDG) can be utilized as alternative protein sources for pet food containing 25 to 43% crude protein [11,12]. These co-products have a higher protein content compared to the initial grain due to the conversion of starch to ethanol as well as the residual yeast component incorporated during processing. DDGS and DDG also have a fiber component ranging from 30 to 55% of total dietary fiber [13,14]. This fiber component, in addition to the yeast cell wall, may serve as a substrate for microbial fermentation in the gut, acting as a prebiotic. For example, previous studies have reported that mannan-oligosaccharides (MOS), an extract from yeast cell walls, support gut health in dogs [15,16]. Therefore, the inclusion of DDGS and DDG into pet food could support intestinal health through modulation of the microbiome and the production of SCFAs [17].

A few studies have evaluated the effects of DDGS on diet digestibility and stool quality in dogs [13,18,19]. Overall, the inclusion of DDGS in canine diets resulted in decreased nutrient digestibility and increased fecal output, likely due to the elevated dietary fiber content. However, there are no studies that have evaluated the effects of traditional DDGS on the microbiome of dogs. Furthermore, there is only one study that has evaluated high-protein DDG (HPDDG; 45% crude protein) and its impact on the canine microbiome [20].

Previous studies have evaluated the effects of corn-fermented protein (CFP), another enhanced DDG, on diet digestibility and stool quality in dogs [19,21,22]. Similar to traditional DDGS, increased inclusion levels of CFP in canine diets resulted in decreased nutrient digestibility and increased fecal output. However, when compared to traditional DDGS, CFP provided greater nutrient utilization and improved stool quality when fed to dogs, which was attributed to the decreased fiber content in CFP compared to DDGS. Corn-fermented protein is produced using post-fermentation technology in which the protein and yeast are separated from the fiber prior to drying. This results in a higher protein, lower fiber ingredient compared to traditional DDGS at 53% crude protein and 35% total dietary fiber [19]. In addition, CFP has a substantial yeast component at approximately 20 to 25% (POET Bioproducts, Sioux Falls, SD, USA). Therefore, the aim of the study was to determine if the heightened yeast component in CFP, compared to traditional DDGS, would promote intestinal health even with decreased fiber content.

## 2. Materials and Methods

Samples for microbial analysis were collected during a digestibility study, which has been previously published [19].

### 2.1. Formulation and Nutritional Composition of the Experimental Diets

The dietary treatments consisted of a control diet containing 15% soybean meal (T1) and experimental diets containing either 3.5% brewer’s dried yeast (T2), 2.5% brewer’s dried yeast plus 17.5% DDGS (T3), or 17.5% CFP (T4). It was assumed that CFP had 20% yeast and DDGS had 5.7% yeast; therefore, all treatments, except T1, were formulated to contain 3.5% yeast. It is important to note that the concentration of yeast in the CFP, DDGS, and dietary treatments was an estimate and not analyzed. In addition, suppliers continue to develop methods to better quantify yeast; therefore, the ingredients may vary in terms of yeast content. The formulated diets met the AAFCO nutritional requirements for adult dogs. The amount of corn, chicken meal, and chicken fat was adjusted between the base rations to maintain nutrient composition among the dietary treatments and resulted in a complete formula (100%). The first base ration was used for T1 and T4 and included all dry ingredients, except for the soybean meal (SBM; Fairview Mills, Seneca, KS, USA), CFP (POET Bioproducts, Sioux Falls, SD, USA), corn starch (Fairview Mills, Seneca, KS, USA), corn gluten meal (CGM; Fairview Mills, Seneca, KS, USA), and titanium dioxide (Fairview Mills, Seneca, KS, USA). The second base ration was used for T2 and T3 and contained all dry ingredients except for SBM, DDGS (Fairview Mills, Seneca, KS, USA), corn starch, CGM, brewer’s dried yeast (BDY; Fairview Mills, Seneca, KS, USA), and titanium dioxide. Soybean meal, CGM, and/or corn starch were added to T1, T2, and T4 to create similar nutrient profiles among all dietary treatments and to balance a 20% inclusion of experimental ingredients compared to T3 (Table 1). 

Each diet was mixed and produced using a single screw extruder (model E525, Extru-Tech, Manhattan, KS, USA). The cool and dry product was packaged in laminated bags and transferred to the laboratory at Kansas State University to be coated. The kibble was coated with chicken fat protected with natural antioxidants (Nutrios, Springfield, MO, USA) and a dry powdered flavor designed for dogs (AFB International, St. Charles, MO, USA). The coated diets were stored in poly-lined Kraft paper bags until fed to the dogs.

The diets were analyzed in duplicate for moisture (AOAC 930.15), ash (AOAC 942.05), fat by acid hydrolysis and hexane extraction (AOAC 960.39), gross energy (Parr 6200 Calorimeter, Parr Instrument Company, Moline, IL, USA), and total dietary fiber (AOAC 991.43). Crude protein was determined by Dumas’ combustion (AOAC 990.03) using a nitrogen analyzer (FP928, LECO Corporation, Saint Joseph, MI, USA). The nutrient composition of the dietary treatments is presented in Table 2.

### 2.2. Feeding Trial

The feeding trial was conducted at the Kansas State University Large Animal Research Center (LARC) under the Institutional Animal Care and Use Committee (IACUC) #4097 protocol.

Twelve healthy adult (6.3 ± 0.45 years) beagles (8 castrated males and 4 spayed females) with an average body weight of 11.4 ± 1.2 kg were individually housed in pens (1.83 m × 1.20 m) equipped with an acrylic-coated mesh floor to allow for separation of urine and feces. The dogs were kept in a temperature-controlled (23 °C) modular building with a 12 h light cycle. The dogs received two feedings per day at 0800 and 1700 h with water ad libitum. The food quantities were determined by calculating the daily metabolizable energy requirement [23] of each dog to maintain body weight. 

### 2.3. Sample Collection

The study was conducted as a replicated 4 × 4 Latin square in which the dogs were randomly assigned to the diets. Each period consisted of 9 days for adaptation followed by 5 days of total fecal collection. During each collection period, a fresh fecal sample from each dog was collected immediately after defecation using a sterile Whirl-Pak bag, and 2 g aliquots were transferred with a spatula into plastic microcentrifuge tubes and stored at −80 °C for DNA extraction.

### 2.4. Fecal DNA Extraction and Sequencing

DNA was extracted from 200 mg of each stool sample (*n* = 48) using a QIAamp Power Fecal Pro DNA Kit (Qiagen, Hilden, Germany) and Qiacube Connect (Qiagen, Hilden, Germany) in accordance with the manufacturer’s instructions (Handbook 02/2020). A nanodrop (NanoDrop 2000, Thermo Scientific, Waltham, MA, USA) was used for quality control of nucleic acid purity. The extractions were quantified on a Qubit fluorometer (Qubit 4.0, Invitrogen by Life Technologies, Carlsbad, CA, USA). The 16S V3/V4 gene was amplified using the Illumina 16S Metagenomic Sequencing library prep protocol (Illumina, Inc., San Diego, CA, USA) as specified by the manufacturer. The size and quality of the libraries and the pool were assessed with a 2100 Bioanalyzer (Agilent, Santa Clara, CA, USA). The libraries were run on an Illumina MiSeq system using 300 × 2 v3 paired-end chemistry.

### 2.5. Data Analysis

Raw reads were trimmed for quality using a CLC Genomics Workbench (Qiagen, v11.0.1) and then imported into mothur (v1.44.1) for further analysis [24]. The unique 16S reads were aligned to reference sequences from the SILVA rRNA database (Release 138) for closed-reference observational taxonomic unit (OTU) assignment [25]. Near-identical sequences were merged using VSEARCH v2.15.1 [26].

Bacterial counts were log-transformed using the log10 command and evaluated using the phyloseq package [27] in R (v4.0.3, R Core Team, 2019). Beta diversity was determined by principal coordinate analysis (PCoA) and alpha diversity was assessed using observed unique sequences, Chao1, Shannon, and Simpson indices. To determine differences among the treatment means, diversity measures were analyzed in a one-way ANOVA in R. Relative abundance data were analyzed within the 50 most abundant OTUs using a GLIMMIX procedure in SAS (v9.4, SAS Institute, Inc., Cary, NC, USA). Tukey’s post hoc test was applied for the least-squares means separation, with significance considered at *p* < 0.05. Treatment was considered the fixed effect, and dog and period were considered as random effects for all statistical analyses. 

## 3. Results

### 3.1. Beta and Alpha Diversity

The PCoA did not provide evidence of clustering among the treatment groups, indicating that the beta diversity was not different among the dietary treatments (Figure 1). In addition, alpha diversity measures were similar (*p* > 0.05) for all dietary treatments (Figure 2).

### 3.2. Phyla Relative Abundance

The predominant phyla within the 50 most abundant observational taxonomic units (OTUs) were Firmicutes (73%), Bacteroidetes (15%), Fusobacteria (8%), and Actinobacteria (4%, Table 3, Figure 3). The relative abundance of Firmicutes was numerically lower for T3 at 69% compared to the remaining treatments at an average of 74%. The relative abundance of Bacteroidetes was 13% for T1, 14% for T2, and 16% for both T3 and T4. The relative abundance of Fusobacteria was numerically greater for T3 at 11% compared to the remaining treatments at an average of 8%. The relative abundance of Actinobacteria was numerically lower for T4 at 3% compared to the remaining treatments at an average of 5%. However, the shifts in the predominant phyla among the dietary treatments were not significant (*p* > 0.05). 

### 3.3. Genera Relative Abundance

Among the 50 most abundant OTUs, the relative abundance of nine genera resulted in differences among the dietary treatments (Table 4). The relative abundance of *Blautia* was greater (*p* < 0.05) for T1 at 12.4% compared to T3 and T4 at an average of 9.1%, with T2 intermediate at 10.4%. The opposite was observed in *Candidatus Stoquefichus* with a greater (*p* < 0.05) relative abundance for T3 and T4 at an average of 1.7% compared to T1 at 0.4%; T2 was intermediate at 1.0%. A decreased (*p* < 0.05) relative abundance in *Collinsella* was observed for T4 at 2.4% compared to T1 and T2 at an average of 3.7%; T3 was intermediate at 3.2%. The relative abundance of *Erysipelatoclostridium* was greatest (*p* < 0.05) for T4 at 5.4% and lowest (*p* < 0.05) for T1 at 1.2%; T3 and T4 were intermediate at 3.8 and 4.7%, respectively. *Peptoclostridium’s* relative abundance was greater (*p* < 0.05) for T2 at 16.6% compared to T1 at 12.5% with T3 and T4 intermediate at an average of 15.2%. The relative abundance of *Phascolarctobacterium* was greater (*p* < 0.05) for T4 at 0.8% compared to T1 at 0.4% with T2 and T3 intermediate at 0.6%. *Romboutsia’s* relative abundance was greater (*p* < 0.05) for T3 at 8.3% compared to T1 at 4.6% with T2 and T4 intermediate at an average of 6.6%. The relative abundance of *Streptococcus* was greater (*p* < 0.05) for T1 at 9.7% compared to T3 and T4 at an average of 0.4%; T2 was intermediate at 5.5%. *Terrisporobacter’s* relative abundance was lower (*p* < 0.05) for T2 at 0.4% compared to T3 at 1.4%, with T1 and T4 intermediate at an average of 0.9%. 

## 4. Discussion

### 4.1. Beta and Alpha Diversity

One of the leading indicators of a healthy gut microbiome is the increased richness and diversity of microorganisms [28]. Dogs with gastrointestinal disorders have been reported to have lower diversity when compared to healthy dogs [29,30,31,32]. Therefore, the preservation of beta and alpha diversity for dogs fed T4 indicates that CFP supported a healthy microbiome compared to traditional ingredients.

A previous study evaluated four diets containing increasing concentrations (0, 7, 14, and 21%) of HPDDG, in exchange for SBM, on the fecal microbiota of dogs [20]. In contrast to the current study where no differences in alpha diversity were observed, the previous study reported a linear increase in the number of OTUs, a quadratic effect for the Shannon index, and a trend for a linear increase in the Chao1 index with increased inclusion of HPDDG [20]. This discrepancy is at least in part due to the variation in dietary fiber content between the historical study and the current study [20]. In addition, the variation in the experimental ingredients among studies could affect the results.

### 4.2. Phyla Relative Abundance

The most abundant phyla in the current study are supported by previous studies, which identified that most bacterial sequences in the canine gastrointestinal tract belong to the phyla Firmicutes, Bacteroidetes, Fusobacteria, Actinobacteria, and Proteobacteria [32,33,34]. For example, Firmicutes has been reported to be the most abundant phylum in fecal samples of healthy dogs at 69% [32], which aligns with the current study at 73%. 

In addition to reduced richness and diversity, the microbiome of dogs in disease states has been characterized by marked shifts in the relative abundance of phyla; specifically, a decrease in Bacteroidetes and Firmicutes and an increase in Actinobacteria [29,35,36,37]. Therefore, a similar relative abundance of phyla among dietary treatments indicates that CFP did not negatively shift the fecal microbiota of dogs compared to traditional ingredients.

The relative abundance of phyla can also be impacted by dietary substrates available for microbial fermentation. It was expected that the dietary treatment with the greatest fiber content (T3) would result in an increase in Firmicutes [7,8,9]. Therefore, the numerical decrease in the relative abundance of Firmicutes in fecal samples of dogs fed T3 was surprising. An increase in dietary fiber has also been reported to increase the production of SCFAs [7,8,9]. However, no significant differences were observed in total SCFA concentrations in the fecal samples of dogs fed the dietary treatments [19]. In contrast, the concentration of propionate was significantly lower in fecal samples of dogs fed T3 compared to dogs fed T1 [19], aligning with the numerical decrease in Firmicutes with T3. In addition, the increased protein content in T1 and T2 would be expected to increase the relative abundance of Fusobacteria [10] whereas the opposite was observed as the relative abundance of Fusobacteria was numerically greater in fecal samples of dogs fed T3. Protein fermentation has also been correlated with the production of branched-chain fatty acids (BCFAs) [38]. However, no significant differences in total BCFAs were observed in the fecal samples of the dogs fed the dietary treatments [19]. The contradictory results in the current study could indicate that the variation in dietary nutrient composition due to the experimental ingredients was not enough to significantly alter the microbial composition at the phyla level.

### 4.3. Genera Relative Abundance

The presence and relative abundance of bacterial genera can provide a more in-depth understanding of the microbial community. Similar to the shifts in the relative abundance of phyla, the presence or increased/decreased abundance of certain genera has been associated with disease. In addition, specific functions of some genera have been reported. Therefore, the type of dietary substrate available for microbial fermentation can be determined, providing insight into the end products of microbial fermentation, such as SCFAs.

The *Blautia* genus has been reported to ferment many types of carbohydrates [39] and improve gut functionality [29,31,40]. The decreased relative abundance of *Blautia* in T3 and T4 was surprising, as the increased dietary fiber content would be expected to increase carbohydrate fermenting microbes. The decrease in *Blautia*, a SCFA-producing bacteria [41], could help to explain the decreased concentration of propionate in fecal samples of dogs fed T3 [19]. In contrast with the current study, *Blautia* increased with HPDDG’s inclusion in the dog’s diets [20]. Similar to alpha diversity, the differing results among studies could be due to the larger variation in fiber content among the dietary treatments in the previous study. 

The genus *Candidatus Stoquefichus* has been previously associated with healthy dogs and is classified as a commensal bacterium [35,36,42,43]. In addition, a reduction in *Candidatus Stoquefichus* in dogs with inflammatory bowel disease (IBD) compared to healthy dogs has been reported [32]. Therefore, the increased relative abundance of *Candidatus Stoquefichus* for T3 and T4 compared to T1 could indicate a beneficial shift.

An increase in the *Collinsella* genus has been observed in dogs with gastric dilation-volvulus (GDV), an acute life-threatening condition [44]. *Collinsella* has also been associated with autoimmune diseases in humans and humanized mice [45]. In addition, the abundance of *Collinsella* has been correlated with the production of proinflammatory cytokine IL-17A, as well as the alteration of gut permeability and disease severity [45]. Therefore, the decrease in *Collinsella* for T4 compared to T1 and T2 may result in a healthier fecal microbiota. 

The genus *Erysipelatoclostridium* has been identified as a possible biomarker for major intestinal diseases such as Crohn’s disease and *Clostridium difficile* infection in humans [46]. In addition, an increase in *Erysipelatoclostridium* was observed for dogs infected with canine parvovirus compared to healthy dogs [47]. However, a significant reduction of *Erysipelatoclostridium* in dogs with acute diarrhea has also been reported [48]. *Erysipelatoclostridium* is known to produce acetate and lactate by metabolizing proteins [49]. Therefore, it could be expected that the treatments with the greatest protein content (T1 and T2) would result in a greater relative abundance of *Erysipelatoclostridium.* In the current study, the opposite response was observed. In addition, the concentration of acetate in the fecal samples of the dogs fed the dietary treatments did not differ [19].

An increase in *Peptoclostridium* has been related to obesity, metabolic syndrome, acute diarrhea, and IBD [50,51,52,53]. Therefore, the combination of distiller’s grains and yeast (T3 and T4) resulted in a more desirable abundance of *Peptoclostridium* than yeast alone (T2).

A previous study observed a decrease in the relative abundance of *Phascolarctobacterium* in dogs with IBD [32]. Therefore, the increase in *Phascolarctobacterium* for T4 compared to T1 could indicate that CFP promoted intestinal health. *Phascolarctobacterium* has been shown to be related to the synthesis of the SCFA propionate [54]. However, the propionate concentration in the fecal samples of the dogs fed the dietary treatments was similar for T1 and T4 [19]. A previous study reported a reduction in *Phascolarctobacterium* after dogs were fed a high-protein diet compared to a medium- or low-protein diet [55]. Therefore, the increased protein content in T1 compared to T4 could have impacted the relative abundance of *Peptoclostridium* in the current study.

*Romboutsia* has a broad range of metabolic capabilities including the utilization of carbohydrates and the fermentation of amino acids [56]. A previous study reported a decrease in *Romboutsia* when dogs were shifted to raw diets from kibble diets, indicating that *Romboutsia* might play an important role in carbohydrate utilization in the hindgut of dogs [57]. Therefore, the increased fiber content in T3 may have resulted in a significant increase in *Romboutsia* for T3 compared to T1. In addition, an increase in *Romboutsia* has been correlated with an increase in acetic, butyric, and propionic acids [58]. However, in the current study, acetate and butyrate concentrations were maintained in the fecal samples of the dogs fed T3 compared to T1. Furthermore, the concentration of propionate in the fecal samples was lower for the dogs fed T3 compared to T1 [19].

The *Streptococcus* genus has been classified as a heterofermentative bacterium that can produce lactic acid and has been previously associated with dogs with IBD [29,32,35,36,59]. Furthermore, *Streptococcus* overgrowth has been considered a hallmark of canine dysbiosis [36,40,60]. Therefore, the decreased relative abundance of *Streptococcus* for T3 and T4 compared to T1 could be beneficial. A decrease was also reported in the genus *Streptococcus* when HPDDG was fed to dogs [20]. 

Previous studies have suggested that *Terrisporobacter* is a pathogenic bacterium [61,62,63,64]. For example, it was concluded that *Terrisporobacter* might regulate enzymes in bile acid metabolism or lipid biosynthesis, eventually leading to higher serum lipid levels and dyslipidemia [64]. *Terrisporobacter* has also been positively correlated with oxidative stress in humans and mice [63,65]. Therefore, a low relative abundance of *Terrisporobacter* would be beneficial. In addition, a negative correlation of *Terrisporobacter* with propionate and total SCFA production has been reported [63]. The highest relative abundance of *Terrisporobacter* was observed in T3 which correlated with a reduction in propionate. However, total SCFA concentrations were not impacted [19].

Taken together, the differences in the relative abundance of genera for the dogs fed the experimental treatments in this study indicate a healthier shift of the fecal microbiota compared to the dogs fed the control. Therefore, the use of co-products in animal nutrition could potentially improve animal health, which is supported by a study conducted in pigs [66]. In addition to potential health benefits, the utilization of co-products as alternative protein sources could provide more sustainable products. This is especially important in pet food, as current protein sources can directly compete with the human food supply. 

## 5. Limitations

Possible limitations of this study could be the lack of yeast quantification, small sample size, and short study duration. It would have been more accurate to analyze the yeast composition in the experimental ingredients and dietary treatments rather than using estimated values. In addition, it would have been beneficial to collect and analyze multiple fecal samples throughout the study for comparison. Also, an increase in the number of dogs enrolled in the study would provide more evidence of the potential impact of CFP on the fecal microbiota. For any study evaluating the impact of dietary intervention, there is concern regarding the study’s duration and if it is long enough for adaptation. However, a study reported that the microbiome of dogs stabilized 6 days after dietary intervention [67], indicating that the 9-day adaptation in the current study should have been sufficient. Of note, the methods in the current study are similar to those previously utilized to evaluate the impact of dietary intervention on the fecal microbiota of dogs.

## 6. Conclusions

In conclusion, CFP did not alter the overall diversity of the fecal microbiota of healthy adult dogs over a 14 d period based on the maintenance of alpha diversity and beta diversity and the relative abundance of phyla compared to traditional ingredients. However, experimental ingredients did shift the canine fecal microbiota at the genus level, likely due to the variation in dietary fiber content. The unique combination of fiber and yeast in CFP appeared to promote the intestinal health of dogs when compared to SBM based on a decrease in genera associated with disease and an increase in genera reported to be commensal and/or promote intestinal health. Future research to determine the potential impact of the yeast component of CFP on additional animal health parameters, such as the immune response, would be valuable. 

## Figures and Tables

**Figure 1 vetsci-10-00553-f001:**
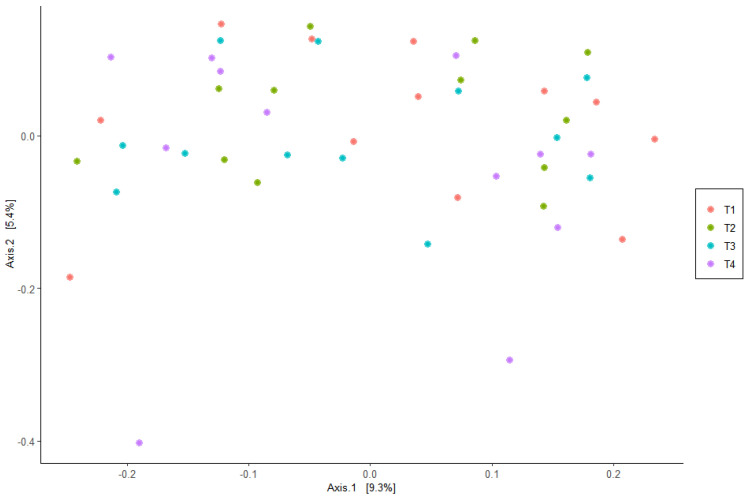
Principal coordinate analysis (PCoA) explaining 9.3% and 5.4% of the variability in operational taxonomic units (OTUs) of Bray–Curtis UniFrac distances for fecal samples from the dogs fed the dietary treatments. Treatments: T1 = control; T2 = brewer’s dried yeast; T3 = brewer’s dried yeast and distiller’s dried grains with solubles; T4 = corn-fermented protein.

**Figure 2 vetsci-10-00553-f002:**
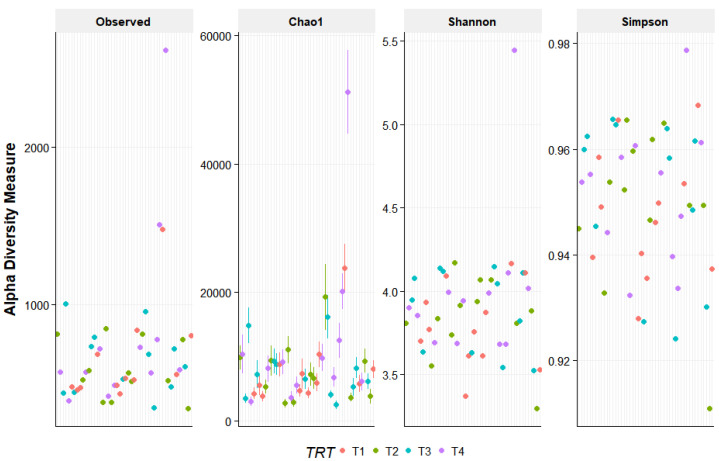
Alpha diversity measures for fecal samples from the dogs fed the dietary treatments. Treatments: T1 = control; T2 = brewer’s dried yeast; T3 = brewer’s dried yeast and distiller’s dried grains with solubles; T4 = corn-fermented protein.

**Figure 3 vetsci-10-00553-f003:**
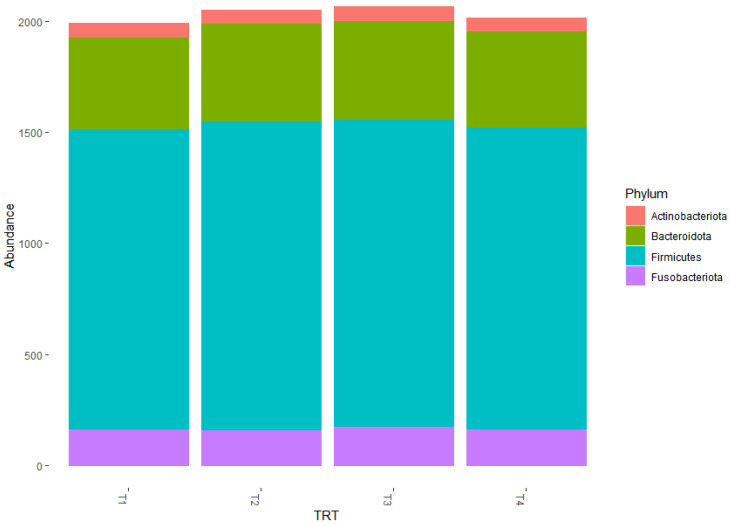
Graphical representation for the relative abundance of bacterial phyla among the 50 most abundant operational taxonomic units (OTUs) in fecal samples from the dogs fed the dietary treatments. Treatments: T1 = control; T2 = brewer’s dried yeast; T3 = brewer’s dried yeast and distiller’s dried grains with solubles; T4 = corn-fermented protein.

**Table 1 vetsci-10-00553-t001:** Ingredient composition of canine diets containing yeast and ethanol co-products on an as-is basis.

	Treatment ^1^			
Ingredient, %	T1	T2	T3	T4
Corn	34.6	30.0	30.0	34.6
Chicken meal	30.0	35.0	35.0	30.0
Soybean meal	15.0	8.0	-	-
Distiller’s dried grains with solubles	-	-	17.5	-
Corn-fermented protein	-	-	-	17.5
Brewer’s dried yeast	-	3.5	2.5	-
Corn starch	-	6.5	-	2.5
Corn gluten meal	5.0	2.0	-	-
Chicken fat	6.0	5.6	5.6	6.0
Other ^2^	9.4	9.4	9.4	9.4

^1^ T1, control; T2, brewer’s dried yeast; T3, brewer’s dried yeast and distiller’s dried grains with solubles; T4, corn-fermented protein. ^2^ Other ingredients: beet pulp, fish meal, flavor, titanium dioxide, salt, potassium chloride, vitamin and mineral premix, choline chloride, and natural antioxidants.

**Table 2 vetsci-10-00553-t002:** Analyzed chemical composition of canine diets containing yeast and ethanol co-products on a dry matter basis.

	Treatment ^1^			
Nutrient	T1	T2	T3	T4
Dry matter, %	95.61	95.92	94.78	95.38
Organic matter, %	90.54	90.44	90.62	91.78
Ash, %	9.46	9.56	9.38	8.22
Crude protein, %	41.13	40.82	38.18	37.55
Fat, %	13.15	13.07	14.82	13.70
Total dietary fiber, %	13.58	13.16	18.39	15.07
Insoluble dietary fiber, %	10.03	10.02	14.28	12.41
Soluble dietary fiber, %	3.65	3.14	4.10	2.64
Gross energy, kcal/kg	5008.71	4988.17	5073.11	5054.00

^1^ T1, control; T2, brewer’s dried yeast; T3, brewer’s dried yeast and distiller’s dried grains with solubles; T4, corn-fermented protein.

**Table 3 vetsci-10-00553-t003:** Relative abundance of bacterial phyla among the 50 most abundant operational taxonomic units (OTUs) in fecal samples from the dogs fed the dietary treatments.

	Treatment ^1^					
Phylum, %	T1	T2	T3	T4	SEM	*p*-Value
Firmicutes	74.59	74.12	69.01	72.40	3.278	0.3310
Bacteroidetes	13.12	14.05	15.78	15.98	2.615	0.6459
Fusobacteria	7.34	7.67	10.60	8.34	1.379	0.1003
Actinobacteria	4.96	4.17	4.61	3.28	0.861	0.2563

^1^ T1, control; T2, brewer’s dried yeast; T3, brewer’s dried yeast and distiller’s dried grains with solubles; T4, corn-fermented protein.

**Table 4 vetsci-10-00553-t004:** Relative abundance of bacterial genera among the 50 most abundant operational taxonomic units (OTUs) in fecal samples from the dogs fed the dietary treatments.

	Treatment ^1^					
Genus, %	T1	T2	T3	T4	SEM	*p*-Value
*Allobaculum*	3.35	2.54	1.63	2.76	0.791	0.2032
*Alloprevotella*	1.18	1.73	1.35	1.33	0.496	0.7216
*Anaerovoracaceae ge*	0.57	0.36	0.55	0.65	0.176	0.4185
*Bacteroides*	9.30	9.78	11.65	11.06	1.974	0.6110
*Bifidobacterium*	1.15	0.54	1.38	0.87	0.729	0.6881
*Blautia*	12.42 ^a^	10.44 ^a,b^	8.88 ^b^	9.35 ^b^	0.984	0.0056
*Candidatus Stoquefichus*	0.42 ^b^	1.04 ^a,b^	1.75 ^a^	1.58 ^a^	0.355	0.0032
*Catenibacterium*	0.66	0.36	0.50	0.48	0.200	0.5220
*Clostridium sensu stricto 1*	0.87	1.04	1.37	1.25	0.324	0.4392
*Collinsella*	3.81 ^a^	3.62 ^a^	3.23 ^a,b^	2.41 ^b^	0.407	0.0086
*Dubosiella*	1.38	0.37	1.89	1.62	1.443	0.7407
*Erysipelatoclostridium*	1.15 ^c^	3.80 ^b^	4.72 ^a,b^	5.40 ^a^	0.533	<0.0001
*Erysipelotrichaceae UCG-003*	1.95	1.58	1.15	2.08	0.666	0.5151
*Faecalibacterium*	2.56	2.55	2.50	3.14	0.524	0.5708
*Faecalibaculum*	0.56	0.61	1.83	2.40	0.824	0.0807
*Fusobacterium*	7.34	7.67	10.60	8.34	1.379	0.1003
*Holdemanella*	3.85	3.39	3.65	2.72	0.648	0.3384
*Lachnospiraceae ge*	0.95	1.35	1.12	0.94	0.221	0.2223
*Lachnospiraceae unclassified*	8.17	7.92	7.26	8.14	0.677	0.5131
*Lactobacillus*	0.86	0.94	0.02	2.09	1.355	0.5092
*Peptoclostridium*	12.53 ^b^	16.62 ^a^	15.32 ^a,b^	15.09 ^a,b^	1.388	0.0425
*Peptococcus*	0.57	0.57	0.35	0.30	0.157	0.2022
*Peptostreptococcus*	1.23	0.00	0.20	0.00	0.768	0.3394
*Phascolarctobacterium*	0.36 ^b^	0.59 ^a,b^	0.59 ^a,b^	0.76 ^a^	0.141	0.0662
*Prevotella 9*	2.05	2.03	2.15	3.17	0.920	0.5496
*Prevotellaceae Ga6A1 group*	0.59	0.51	0.63	0.42	0.284	0.8883
*Romboutsia*	4.60 ^b^	6.16 ^a,b^	8.28 ^a^	7.12 ^a,b^	1.096	0.0160
*Streptococcus*	9.69 ^a^	5.47 ^a,b^	0.13 ^b^	0.70 ^b^	2.200	0.0002
*Terrisporobacter*	0.44 ^a,b^	0.39 ^b^	1.35 ^a^	1.32 ^a,b^	0.345	0.0077
*Turicibacter*	5.44	6.02	3.98	2.51	1.320	0.0499

^1^ T1, control; T2, brewer’s dried yeast; T3, brewer’s dried yeast and distiller’s dried grains with solubles; T4, corn-fermented protein. ^a–c^ Means within a row lacking a common superscript letter are different (*p* < 0.05).

## Data Availability

The data presented in this study are openly available in SRA at https://www.ncbi.nlm.nih.gov/bioproject/986920, reference number PRJNA986920.

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
