# Peer review of "Comparison of the Effect of Corn-fermented Protein and Traditional Ingredients on the Fecal Microbiota of Dogs"

_vetsci, 2023, doi:10.3390/vetsci10090553_

Round 1

Reviewer 1 Report

The manuscript of Kilburn-Kappeler et al. reports the impact of an alternative feed on the faecal microbiota of dogs. It is well-known that animal nutrition can influence bacterial composition along the gastrointestinal tract and faeces, thus representing a tool for animal health evaluation. Nevertheless, the manuscript presents several lacks. In my humble opinion, the manuscript cannot be published in its present form. However, I provide some comments and suggestions for the authors to improve the manuscript whether they try to submit again it. 

The title should be revised. In my opinion, it is not appropriate stating that a corn fermented protein supplemented diet was evaluated because the authors tested four different diets on beagle dogs. A suggested one could be “Effect of different dietary treatments on the faecal microbiota on beagle dogs breed”; or, better, another one could be “Comparison between a corn-fermented-protein-supplemented diet with traditional canine diets on the faecal microbiota on beagle dogs breed”.   Lines 23 and 24 can be removed or better moved to the simple summary because of repetition in the sentence with lines 17 and 18. I suggest integrating lines 23 and 24 in the simple summary text.   In the abstract section, no need to provide the version of the software used for analysis (that should be reported just in the M&M section). On the contrary, authors should mention in the abstract at least the most abundant phyla or genera encountered in their analysis.   Keywords could be revised as follows canine diets; corn fermented protein; dried yeast; 16S metagenomics sequencing; faecal microbiota.   In the text, I would avoid using the term “faecal microbiome”, generally used for describing the entire microbial community (bacteria, fungi, viruses etc.) associated with their functional genes. In this study, the authors potentially addressed their focus on the potential shifts of the bacterial community in dogs’ faeces after different dietary treatments. So, I believe that it is appropriate the adoption of the term faecal microbiota.    The introduction section is too short. It would seem more like an Abstract than an introduction. If there is no available information regarding feed integration in dogs, it can be improved using studies performed on other species. Then, authors can reformulate the introduction section to make the manuscript more attractive. In addition, I would avoid some lumps in the citations. It could be better to describe each study reported to provide further detailed information, thus extending the introduction.   In the experimental design section (Feeding Trial and Sample Collection), I ask why the authors did not collect more faecal samples. In the sample collection, the authors stated that the trial lasted just five days; and faeces were collected within 15 minutes and placed in sterile containers. Why such a delay in collecting the faecal samples? It can invalidate sampling quality because faecal collection needs to be performed immediately to preserve the bacterial composition. In addition, in dogs is very simple to monitor and predict defecation and then collection directly from the rectal ampoule.    Regarding the faecal DNA Extraction and Sequencing section, the authors do not specify which regions they sequenced. Considering that they report Libraries were run by Illumina Miseq system using 300x2 v3 paired, I suppose that they investigated a specific region of the 16S gene, even though they do not specify which ones.   Also, the data analysis section presents several lacks. Which statistical tests have been used to estimate the alpha and beta diversity indices? What is the usefulness of applying a mixed-model to estimate the abundance when authors state that there was no significance between or within groups?  It can be more helpful to report an appropriate statistical model in the M&M section. Why do authors state the period as random since they considered just one period? In addition, regarding the use of SAS software, which PROCs have been used? This information is also missing.   Why authors did not perform a differential expression analysis using the DSeq2 package in RStudio? Although the authors affirm that they did not observe significant variations in alpha- and beta-diversity, a differential analysis (e.g., at genus level) could help to understand if some genera were differentially expressed, positively or negatively modulated (e.g., reporting the Log2FC) by the different dietary treatments. I encourage the authors to perform this analysis to provide further potential information in the manuscript. In addition, why the authors did not perform a functional prediction analysis?   To provide more details, it could be better if authors complete this subsection with information regarding the packages (with relative citations) used in RStudio. What packages in R were used for alpha diversity estimation? The authors should report the statistical method applied (e.g., Wilcoxon or Kruskal-Wallis tests, and so on) to verify differences among groups. The same for Beta-diversity; authors should report the package used in R.   As for Beta- and Alpha-diversity, also the relative abundance of phyla and genera should be graphically reported. It is better to add some bar plots in the manuscript to illustrate the phyla and genera distribution comparing each treatment.   Finally, in the discussion section, I observed several speculations. If the authors consider resubmitting this manuscript, I suggest revising this section according to the results reported avoiding speculation. I hope the authors do not desist from revising their study.

Author Response

The manuscript of Kilburn-Kappeler et al. reports the impact of an alternative feed on the faecal microbiota of dogs. It is well-known that animal nutrition can influence bacterial composition along the gastrointestinal tract and faeces, thus representing a tool for animal health evaluation. Nevertheless, the manuscript presents several lacks. In my humble opinion, the manuscript cannot be published in its present form. However, I provide some comments and suggestions for the authors to improve the manuscript whether they try to submit again it. 

We greatly appreciate your review and understand your concerns. We hope our revision based on your comments is satisfactory for publication.

The title should be revised. In my opinion, it is not appropriate stating that a corn fermented protein supplemented diet was evaluated because the authors tested four different diets on beagle dogs. A suggested one could be “Effect of different dietary treatments on the faecal microbiota on beagle dogs breed”; or, better, another one could be “Comparison between a corn-fermented-protein-supplemented diet with traditional canine diets on the faecal microbiota on beagle dogs breed”.  

Changed title to “Comparison of corn fermented protein to traditional ingredients on the fecal microbiota of dogs.”

Lines 23 and 24 can be removed or better moved to the simple summary because of repetition in the sentence with lines 17 and 18. I suggest integrating lines 23 and 24 in the simple summary text.  

We think it is beneficial written as is so the abstract can stand alone. If we remove this sentence from the abstract, it is not clear what CFP is.

In the abstract section, no need to provide the version of the software used for analysis (that should be reported just in the M&M section).

This has been removed from the abstract.

On the contrary, authors should mention in the abstract at least the most abundant phyla or genera encountered in their analysis.  

Added “Predominant phyla among all samples were Firmicutes (73%), Bacteroidetes (15%), Fusobacteria (8%), and Actinobacteria (4%).”

Keywords could be revised as follows canine diets; corn fermented protein; dried yeast; 16S metagenomics sequencing; faecal microbiota.  

Keywords have been revised as suggested.

In the text, I would avoid using the term “faecal microbiome”, generally used for describing the entire microbial community (bacteria, fungi, viruses etc.) associated with their functional genes. In this study, the authors potentially addressed their focus on the potential shifts of the bacterial community in dogs’ faeces after different dietary treatments. So, I believe that it is appropriate the adoption of the term faecal microbiota.   

“fecal microbiome” has been changed to “fecal microbiota” throughout the manuscript.

The introduction section is too short. It would seem more like an Abstract than an introduction. If there is no available information regarding feed integration in dogs, it can be improved using studies performed on other species. Then, authors can reformulate the introduction section to make the manuscript more attractive. In addition, I would avoid some lumps in the citations. It could be better to describe each study reported to provide further detailed information, thus extending the introduction.

Additional paragraphs discussing the microbiome have been added to the introduction. Overall outcomes from studies mentioned in the introduction have also been added.

In the experimental design section (Feeding Trial and Sample Collection), I ask why the authors did not collect more faecal samples. In the sample collection, the authors stated that the trial lasted just five days; and faeces were collected within 15 minutes and placed in sterile containers. Why such a delay in collecting the faecal samples? It can invalidate sampling quality because faecal collection needs to be performed immediately to preserve the bacterial composition. In addition, in dogs is very simple to monitor and predict defecation and then collection directly from the rectal ampoule.   

Dogs were fed diets for 9 days prior to the 5-day collection. Each period consisted of 14 days, with 4 periods is a total study duration of 56 days. This study was part of a digestibility trial in which all feces were collected during those 5 days. A fresh fecal sample was collected for each dog in each period during the sample collection phase resulting in 48 total samples. A fecal sample was considered “fresh” when defecated within 15 minutes. However, fecal samples were collected immediately after defecation which usually occurred in the morning after feeding. This section has been reworded for clarification.

Regarding the faecal DNA Extraction and Sequencing section, the authors do not specify which regions they sequenced. Considering that they report Libraries were run by Illumina Miseq system using 300x2 v3 paired, I suppose that they investigated a specific region of the 16S gene, even though they do not specify which ones.  

It was previously stated that the V3/V4 gene was amplified. This section has been reworded for clarification.

Also, the data analysis section presents several lacks. Which statistical tests have been used to estimate the alpha and beta diversity indices?

Additional details have been added to the Data Analysis section.

What is the usefulness of applying a mixed-model to estimate the abundance when authors state that there was no significance between or within groups?  It can be more helpful to report an appropriate statistical model in the M&M section. Why do authors state the period as random since they considered just one period? In addition, regarding the use of SAS software, which PROCs have been used? This information is also missing.  

We selected to use a mixed model to account for both the fixed and random effects in the model. We are not necessarily interested in the effects of dog and period, but it is best practice to include them in the model to account for potential effects. This is the same model we have used in the digestibility trial which has been accepted for publication in Journal of Animal Science. Also, 4 periods were analyzed not 1, which is why we included period as a random effect. Additional details have been added to this section for clarification.

Why authors did not perform a differential expression analysis using the DSeq2 package in RStudio? Although the authors affirm that they did not observe significant variations in alpha- and beta-diversity, a differential analysis (e.g., at genus level) could help to understand if some genera were differentially expressed, positively or negatively modulated (e.g., reporting the Log2FC) by the different dietary treatments. I encourage the authors to perform this analysis to provide further potential information in the manuscript.

The analysis performed in this study aligns with previous microbiome studies in dogs. Specifically, it is similar to the only other study which has evaluated the effects of distillers dried grains on the fecal microbiota of dogs, allowing for a direct comparison. In addition, this manuscript provides more detail than some published papers on this topic.

In addition, why the authors did not perform a functional prediction analysis?  

The analysis performed in this study aligns with previous microbiome studies in dogs. Specifically, it is similar to the only other study which has evaluated the effects of distillers dried grains on the fecal microbiota of dogs, allowing for a direct comparison. In addition, this manuscript provides more detail than some published papers on this topic.

To provide more details, it could be better if authors complete this subsection with information regarding the packages (with relative citations) used in RStudio. What packages in R were used for alpha diversity estimation? The authors should report the statistical method applied (e.g., Wilcoxon or Kruskal-Wallis tests, and so on) to verify differences among groups. The same for Beta-diversity; authors should report the package used in R.  

Additional details have been added to the Data Analysis section.  

As for Beta- and Alpha-diversity, also the relative abundance of phyla and genera should be graphically reported. It is better to add some bar plots in the manuscript to illustrate the phyla and genera distribution comparing each treatment.  

The graphical representation of the relative abundance of phyla has been added. Due to the large number of genera reported we would prefer to leave as is for clarity and efficient use of space.

Finally, in the discussion section, I observed several speculations. If the authors consider resubmitting this manuscript, I suggest revising this section according to the results reported avoiding speculation. I hope the authors do not desist from revising their study.

Discussion sections of microbiome studies often include speculations; therefore, the writing style of this manuscript is not far off. However, direct speculations have been deleted/reworded and more support has been added to the discussion section.

Reviewer 2 Report

It is a very interesting manuscript that provides information on the effect of corn fermented protein compared to traditional ingredients on the fecal microbiome of dogs. The reviewer thanks the authors for the good work and appreciates their contribution to the literature.

See my minimal comments below.

Title

The reviewer suggests avoiding abbreviations in the title, specifically “CFP”.

Abstract

Line 36: Replace “14-d” for “14 days.”

Introduction

Lines 43-44: If the authors are referring to DDGS and DDG in general (all types of grains and not only corn), then these diets contain residual yeast protein in addition to grain protein instead of just corn protein, as they stated. Then I suggest using "grain protein" instead of "corn protein".

Discussion

The authors should consider including some limitations of the study, such as the small sample size.

Author Response

It is a very interesting manuscript that provides information on the effect of corn fermented protein compared to traditional ingredients on the fecal microbiome of dogs. The reviewer thanks the authors for the good work and appreciates their contribution to the literature.

We greatly appreciate your review and are happy to hear that the manuscript was interesting. We hope we have addressed your comments to be satisfactory.

See my minimal comments below.

Title

The reviewer suggests avoiding abbreviations in the title, specifically “CFP”.

“CFP” has been removed from the title.

Abstract

Line 36: Replace “14-d” for “14 days.”

Changed to “over 14 days.”

Introduction

Lines 43-44: If the authors are referring to DDGS and DDG in general (all types of grains and not only corn), then these diets contain residual yeast protein in addition to grain protein instead of just corn protein, as they stated. Then I suggest using "grain protein" instead of "corn protein".

Great point! However, the introduction has been restructured based on another reviewer’s comments and this sentence has been removed.   

Discussion

The authors should consider including some limitations of the study, such as the small sample size.

Thank you for mentioning this. We have added a Limitations section at the end of the discussion.

Reviewer 3 Report

Dear Authors,

Congrats on a nice study.

57

This is the first study to evaluate CFP on the fecal microbiome of dogs – rephrase.

285 - 288

However, experimental ingredients did shift the canine microbiome on a genus level, likely due to the type of substrate in the gastrointestinal tract. The unique combination of fiber and yeast in CFP may promote the intestinal health of dogs when compared to SBM.  conclusion seems a bit vague – please use some firm conclusions from the discussion to generate a more concrete conclusion for your study.

Was there a specific reason why the protein in the diet was on the high side and the fat level was on the low end for commercial dog adult diets? Could that have an influence on your results?

Study limitations - such as yeast quantification, etc., and duration of the study (is it long enough to have substantial changes in the microbiome) should be addressed at the end of the discussion alongside recommendations for further research.

Kindest regards,

Author Response

Congrats on a nice study.

Thank you for your review. We are glad to hear that the study was presented well. We hope we have addressed your comments to be satisfactory. 

57

This is the first study to evaluate CFP on the fecal microbiome of dogs – rephrase.

The introduction has been restructured based on another reviewer’s comments and this sentence has been removed.  

285 - 288

However, experimental ingredients did shift the canine microbiome on a genus level, likely due to the type of substrate in the gastrointestinal tract. The unique combination of fiber and yeast in CFP may promote the intestinal health of dogs when compared to SBM.  – conclusion seems a bit vague – please use some firm conclusions from the discussion to generate a more concrete conclusion for your study.

Evidence has been added to the conclusion to create a more concrete conclusion.

Was there a specific reason why the protein in the diet was on the high side and the fat level was on the low end for commercial dog adult diets? Could that have an influence on your results?

These diets were also fed to cats which required adjustments to the nutrient composition. We also wanted to achieve a similar nutrient profile to a previous study in our lab which evaluated a higher inclusion level of CFP (Smith and Aldrich, 2023). Overall, the general population of the microbiome in this study is similar to those previously reported in dogs. Therefore, the higher protein content and lower fat content for diets overall did not appear to impact microbial composition. In addition, nutrient values are similar to some commercial products on a dry matter basis. However, the scope of this project was to investigate the effects of the various ingredient sources, not necessarily a comparison to commercial products. Regardless, we do understand your point and it would be beneficial to address this in a future study.

Study limitations - such as yeast quantification, etc., and duration of the study (is it long enough to have substantial changes in the microbiome) should be addressed at the end of the discussion alongside recommendations for further research.

Thank you for pointing this out. A Limitations section has been added to the end of the discussion. In addition, a recommendation for further research has been added to the end of the Conclusions section.

Round 2

Reviewer 1 Report

I thank the Authors for following my comments and suggestions. I am glad to see they considered my report for improving the manuscript. However, before publication, I provide just some additional comments. 

At the end of the introduction section, the Authors should clearly state the aim of the study.

Line 32. The names of bacterial species should be reported in Italic style. Please revise this aspect for phyla, genera, and species names in the entire text.

Line 34. The letter p of the p-value should be reported in lowercase in the entire text.

Line 45. Please, report the abbreviation in brackets for short-chain fatty acids in SCFAs, (also for line 63 and so on in the text).

Line 75. Authors should better clarify the abbreviations "Distillers dried grains with (DDGS) or without solubles “proteins” (DDG)". I am not a vet nutritionist, but in my opinion, the distinction with or without solubles is not completely understandable. If the Authors agree with me, this aspect may be revised as follows, DDGs-S or S-DDGs for distillers' dried grains (with solubles) and DDGs for grains without solubles protein. If the Authors agree, they should revise it in the entire manuscript, specifying along the text “solubles protein” where needed.

Please, check the correctness of all abbreviations reported in the whole manuscript (e.g., from line 107 until 149). If already abbreviated, do not need to report in extended form (e.g., line 337).

Line 106. I would suggest to revise the title of paragraph 2.1 as follows, “Formulation and nutritional composition of the experimental diets”.

Line 202. Please, add the following reference for the phyloseq package: 

McMurdie, P.J.; Holmes, S. phyloseq: An R package for reproducible interactive analysis and graphics of microbiome census data. PLoS ONE 20138, e61217.

Paragraph 3.3.

As for paragraph 3.2, also in this section, it should be reported a barplot indicating the relative abundance of genera identified in the samples (e.g., Authors can report only the ten top taxa).

I appreciate if the authors may include in the bibliography of their manuscript the following studies, underlying (e.g., in the discussion section) the importance of the use of alternative feedstuffs in animal nutrition (e.g., also in other species such as pigs) for improving gut microbiota, animal wellness, and growth performance.

Sutera, A.M.; Arfuso, F.; Tardiolo, G.; Riggio, V.; Fazio, F.; Aiese Cigliano, R.; Paytuví, A.; Piccione, G.; Zumbo, A. Effect of a Co-Feed Liquid Whey-Integrated Diet on Crossbred Pigs’ Fecal Microbiota. Animals 2023, 13, 1750. https://doi.org/10.3390/ani13111750

Author Response

I thank the Authors for following my comments and suggestions. I am glad to see they considered my report for improving the manuscript. However, before publication, I provide just some additional comments. 

We appreciate your follow-up comments and hope we have addressed them to be satisfactory.

At the end of the introduction section, the Authors should clearly state the aim of the study.

The last sentence of the introduction has been converted to clearly state the aim of the study.

Line 32. The names of bacterial species should be reported in Italic style. Please revise this aspect for phyla, genera, and species names in the entire text.

It is our understanding that names should only be italicized at the family level and below, therefore phyla names should not be italicized. Genera and species names have been italicized.

Line 34. The letter p of the p-value should be reported in lowercase in the entire text.

Changed to lowercase throughout entire text.

Line 45. Please, report the abbreviation in brackets for short-chain fatty acids in SCFAs, (also for line 63 and so on in the text).

SCFA has been changed to SCFAs when appropriate.

Line 75. Authors should better clarify the abbreviations "Distillers dried grains with (DDGS) or without solubles “proteins” (DDG)". I am not a vet nutritionist, but in my opinion, the distinction with or without solubles is not completely understandable. If the Authors agree with me, this aspect may be revised as follows, DDGs-S or S-DDGs for distillers' dried grains (with solubles) and DDGs for grains without solubles protein. If the Authors agree, they should revise it in the entire manuscript, specifying along the text “solubles protein” where needed.

The current abbreviations follow definitions from AAFCO and the Distillers Grains Technology Council. The abbreviations are also in alignment with other manuscripts. DDGS (with capital S) refers to solubles whereas DDGs (lower case s) refers to distillers dried grains (plural). The solubles are an additional fiber component not protein. Therefore, we would like to leave the abbreviations as is.

Please, check the correctness of all abbreviations reported in the whole manuscript (e.g., from line 107 until 149). If already abbreviated, do not need to report in extended form (e.g., line 337).

Use of abbreviations have been corrected throughout manuscript.

Line 106. I would suggest to revise the title of paragraph 2.1 as follows, “Formulation and nutritional composition of the experimental diets”.

Title has been revised as suggested.

Line 202. Please, add the following reference for the phyloseq package: 

McMurdie, P.J.; Holmes, S. phyloseq: An R package for reproducible interactive analysis and graphics of microbiome census data. PLoS ONE 20138, e61217.

Reference has been added.

Paragraph 3.3.

As for paragraph 3.2, also in this section, it should be reported a barplot indicating the relative abundance of genera identified in the samples (e.g., Authors can report only the ten top taxa).

In our opinion adding a bar plot would be repetitive, since a table providing the relative abundance of genera is already included. In addition, we think the table provides more useful information as the exact values are provided rather than just visual estimates. Therefore, we would prefer not to include a bar plot of the genera.

I appreciate if the authors may include in the bibliography of their manuscript the following studies, underlying (e.g., in the discussion section) the importance of the use of alternative feedstuffs in animal nutrition (e.g., also in other species such as pigs) for improving gut microbiota, animal wellness, and growth performance.

Sutera, A.M.; Arfuso, F.; Tardiolo, G.; Riggio, V.; Fazio, F.; Aiese Cigliano, R.; Paytuví, A.; Piccione, G.; Zumbo, A. Effect of a Co-Feed Liquid Whey-Integrated Diet on Crossbred Pigs’ Fecal Microbiota. Animals 202313, 1750. https://doi.org/10.3390/ani13111750

Additional information (including the suggested reference) has been added to the last paragraph in the Genera Relative Abundance discussion section.